# Subclinical Mastitis Related to *Streptococcus canis* Infection in Dairy Cattle

**DOI:** 10.3390/vetsci12030286

**Published:** 2025-03-19

**Authors:** Alessio Sposato, Laura Del Sambro, Stefano Castellana, Elisabetta Catalano, Michela Galgano, Antonella Castellana, Annamaria Caffò, Viviana Manzulli, Marta Caruso, Leonardo Marino, Angelica Milano, Luciana Addante

**Affiliations:** 1Department of Public Health, Experimental and Forensic Medicine, University of Pavia, Via Carlo Forlanini 2, 27100 Pavia, Italy; alessio.sposato@izspb.it; 2Istituto Zooprofilattico Sperimentale della Puglia e della Basilicata, Via Manfredonia 20, 71121 Foggia, Italy; stefano.castellana@izspb.it (S.C.); elisabetta.catalano@izspb.it (E.C.); michela.galgano@izspb.it (M.G.); antonella.castellana@izspb.it (A.C.); annamaria.caffo@izspb.it (A.C.); viviana.manzulli@izspb.it (V.M.); marta.caruso@izspb.it (M.C.); leonardo.marino@izspb.it (L.M.); angelica.milano@izspb.it (A.M.); luciana.addante@izspb.it (L.A.)

**Keywords:** *Streptococcus canis*, bacteria, bovine mastitis, cattle, whole genome sequencing, intramammary infection

## Abstract

Mastitis is described as an inflammatory process affecting the mammary gland and is a major problem in dairy cattle. Infectious causes are among the most common in the development of inflammatory processes. Pathogens most commonly involved include bacteria. Among these, *Streptococcus canis* could represent a contagious pathogen causing subclinical mastitis. This study reports a case of subclinical mastitis in a dairy farm where the hygienic conditions were moderate and free domestic carnivores were present. The positive animals were treated, and another sampling was performed for the positive animals and for the rest of the lactating ones. Swabs samples from the pharyngeal mucosa of the dogs were performed in order to identify the possible source of infection. The strong selective pressure exerted by the improper use of antibiotics has allowed for the development of bacterial strains resistant to the common drugs used. Antibiotic resistance represents an increasingly present problem in the world. This aspect, together with the possibility of some bacteria to cause disease in animals and humans, pushes for greater awareness of the risks associated with the development of resistant strains in livestock farming.

## 1. Introduction

*Streptococcus* (*S.*) *canis* is a Gram-positive cocci, catalase, and CAMP-negative bacterium, described as β-hemolytic and belonging to the pyogenic Lancefield group G Streptococcus. Originally isolated from dogs and cows in 1986 by Devriese [1] and included in the Lancefield group G antigen (GGS), *S. canis* is normally detected in the respiratory tract of dogs and cats, but it can also be considered an opportunistic pathogen of different tissues, such as the central nervous system, genitourinary system, blood, skin, bones, and cardiovascular system [1,2,3,4]. The zoonotic pathogen has also been isolated in human genital, urinary, and upper respiratory tract infections through transmission by dogs [5,6]. Infection by this microorganism can also affect other animals, including minks (*Mustela vison*), mice (*Mus* spp.), rabbits (*Oryctolagus cuniculus*), and foxes (*Vulpes* spp.) [1,7,8], and occasionally can be isolated from the udders or milk of dairy cows [9]. *S. canis* in dairy herds can represent a rare but contagious pathogen causing intramammary infection (IMI) [10,11], being diagnosed in the udder in only 0.7% of all streptococcal infections [12]. Potential sources of infection in cows are cats and dogs with access to the barn [10,13]. Infected cows have macroscopically normal udders (integral skin; normal sphincter size; no evident excoriations, ulcers, vesicles, or morphological alterations), showing an increase in somatic cell counts (SCCs). They can spread the pathogen via the milking procedure, especially when the farm adopts poor udder health management, such as an ineffective post-milking teat disinfection. These factors can lead to the development of massive outbreaks of clinical and subclinical mastitis [13]. Usually, bacterial strains responsible for these outbreaks on farms are attributable to a single clone or closely related on a phylogenetic level [10]. The pathogenesis in cattle is still unclear regarding the bacterium’s ability to establish infection in the udder tissue and the amount of economical loss and damage caused [14,15]. In this paper, we describe a case of bovine subclinical mastitis caused by *S. canis* in an Apulian dairy herd. The source of infection was not identified, but the bacteria identification and characterization with molecular typing of the GGS isolates suggests cow-to-cow transmission.

## 2. Materials and Methods

### 2.1. Animal and Herd Description

In December 2023, a dairy farm located in Apulia (Italy), was inspected for cases of possible subclinical mastitis connected to a reduced level of milk production. The farmer reported that the average milk production decreased from 30 kg/cow per day to 27 kg/cow. Furthermore, no cattle from other farms were introduced into the farm and the average daily milk production was lower than the national average. The herd, consisting of 51 heads of Holstein–Friesian cows, with 31 lactating, was housed in a tie-stall barn with concrete floors on straw beds and deep litter, replaced every 30 days. Animals were fed a total mix ration (TMR) diet. The cows were milked twice daily with a 5.08 cm pipeline around the barn with 8 milking units. The average lactation duration was 290 days, and the average lactation was 3, which reflects the national average. The breeder had not adhered to any facultative breeding plan for cows. Cows were not separated from other ones during the milking procedures. Biosecurity deficiencies in the management of the animals were reported, such as a failure to separate infected animals from healthy ones during milking, and free access of dogs and cats to the cow housing rooms. The management of the herd and the hygienic conditions during the milking process were evaluated: no pre-dipping or post-dipping treatment was performed, cluster flushing was carried out at the end of the total herd milking procedures, and dry cow treatment was not reported. No routine bacteriological examination of the milk was planned.

### 2.2. Sample Collection and Bacteriological Culture and SCCs

During the farm visit, the CMT and a clinical examination (inspection and palpation) of the udder health of each cow was performed on every lactating cow in the herd. Milk samples from each animal that showed high viscosity and gelatinous appearance in the CMT (8 samples) were collected aseptically according to the method described by the National Mastitis Council [16]. Sampling was only carried out of cows in order to ensure sterile conditions during the collection procedure. Sampling was repeated, employing the same specifications, after six months in order to assess the evolution of the infection and the success of therapy (see Section 3.2 for details). Milk samples were stored in cooler boxes with ice packs (2–8 °C) and were transported to the laboratory of the Istituto Zooprofilattico Sperimentale of Apulia and Basilicata placed in Foggia and Putignano for the bacteriological analyses. The SCCs were determined using flow cytometry equipment: a fossomatic cell counter (AS Foss, Hillerod, Denmark). Microbiological procedures were performed according to the National Mastitis Council guidelines [16]. Milk samples (10 µL) were inoculated onto Columbia blood agar (CBA), Mannitol salt agar (MSA), MacConkey agar (MCK), and Modified Edwards medium plates (ED) (Liofilchem, Teramo, Italy) and incubated at 37 °C, for 24–48 h under aerobic conditions. Bacterial cultures were inspected visually and growth of one or more uniform colonies were considered relevant and growth of three and more different colonies were considered as contamination. Bacteria were identified to the genus level according to their colony morphology and hemolytic patterns, Gram stain, catalase test, and biochemical tests. Colonies suspected to be streptococci were tested for the Lancefield group using the latex agglutination test (Liofilchem, Teramo, Italy). Bacterial isolates with doubtful results were identified biochemically using the API System (bioMerieux, Marcy-l’Étoile, France). Then, after the follow-up, the farmyard dogs were clinically examined and swabs from the nasal or pharyngeal mucosa were collected for evaluating the possible role as a contaminant and cultured onto Columbia blood agar (CBA), Mannitol salt agar (MSA), MacConkey agar (MCK), and Modified Edwards medium plates (ED) (Liofilchem, Teramo, Italy) and incubated at 37 °C, for 24–48 h, under aerobic conditions.

### 2.3. MALDI-TOF MS Methodology

Bacterial colonies tested with ambiguous results were subjected to species identification using MS MALDI-TOF mass spectrometry [17]. All isolates were analyzed in duplicate. The data were processed automatically by MBT Compass 4.1.70 software (Bruker Daltonik GmbH, Bremen, Germany) and the mass spectra were compared with those of the commercial database provided by Bruker Daltonics (Milano, Italia).

### 2.4. Antimicrobial Susceptibilities Test

The bacterial isolates were tested for their susceptibility to antibiotics using standardized method according to the National Committee for Clinical Laboratory Standards (NCCLS); in particular, the dilution susceptibility test was performed as previously described by Farina et al. [18]. The following antimicrobials were tested: rifampicin, trimethoprim, tetracycline, gentamicin, ampicillin, penicillin, ceftiofur, cefazolin, enrofloxacin, pirlimycin, erythromycin, and amoxicillin/clavulanic acid. All the plates were incubated at 35 (±2) °C for 24 h at aerobic conditions. The lowest concentration capable of inhibiting the appearance of visible growth of the test microorganism is known as the minimum inhibitory concentration (MIC). MIC is interpreted using a clinical breakpoint for categorizing the microorganism as susceptible (S) or resistant (R) [19].

### 2.5. Diagnostic PCR Analyses

In order to evaluate the pathogen responsible of subclinical mastitis, Real-time PCR (qPCR) analysis was performed with commercial kits. The DNA of each milk sample was extracted using MagMAX^™^ CORE Mastitis & Panbacteria Module kit (Thermo Fisher Scientific, Waltham, MA, USA) according to the manufacturer’s protocol; then, the Mutiplex qPCR assay was performed with the VetMAX^™^ MastiType Multi kit (Thermo Fisher Scientific, Waltham, MA, USA). The kit allows for the detection of 15 possible mastidogen pathogens and a beta lactamase resistance gene by amplification in four separate reactions (*Staphylococcus aureus*, *Staphylococcus* spp. (including all major coagulase-negative staphylococci), *Streptococcus agalactiae*, *Streptococcus dysgalactiae*, *Streptococcus uberis*, *Escherichia coli*, *Enterococcus* spp. (including *E. faecalis* and *E. faecium*), *Klebsiella oxytoca*, and/or *K. pneumoniae*, *Serratia marcescens*, *Corynebacterium bovis*, *Trueperella pyogenes* and/or *Peptoniphilus indolicus*, Staphylococcal β-lactamase gene (penicillin-resistance gene), *Mycoplasma bovis*, *Mycoplasma* spp., yeast, and *Prototheca* spp.

### 2.6. Genotyping and Genomic Characterization

Genotyping for the assembled genomes was performed by Multi-Locus Sequence Typing on the PubMLST website (https://pubmlst.org/bigsdb?db=pubmlst_scanis_seqdef, accessed on 25 September 2024) [20]. Furthermore, sample-specific Sequence Type was queried onto the PubMLST platform in order to possibly retrieve isolates with analogous genotype and host/source. Species attribution was further confirmed by genome to genome comparison within the JSpecies platform and ANIBlast [21] method (https://jspecies.ribohost.com/jspeciesws/#anib, accessed on 25 September 2024).

## 3. Results

### 3.1. First Isolation

The Californian Mastitis Test was positive for 8 animals, but the udders were normal upon the clinical examination (inspection and palpation), and none showed evidence of clinical mastitis. The SCC was >200 × 10^3^ cell/mL in three cows, respectively, sample 2 (9094 × 10^3^ cell/mL), sample 4 (608 × 10^3^ cell/mL), and sample 7 (3619 × 10^3^ cell/mL), confirming the presence of subclinical mastitis. The milk samples coming from the same three animals were positive on CBA plates after 24 h incubation in aerobic conditions, showing numerous grey-white β-hemolytic smooth colonies with a diameter of 2–3 mm. The same growth was observed in ED plates. No bacterial growth was observed in the remaining samples even after 48 h of incubation in all the media used. Bacteria from the three positive samples were Gram-stained and catalase tests were evaluated; pure Gram-positive cocci catalase-negative colonies were found in all cases. The strains were esculin-negative and CAMP-negative and positive for the Lancefield group G antigen and so they were considered as group G streptococci (GGS). All the three isolates were then identified as *S. canis* by using the API Strep system (bioMerieux, Marcy-l’Étoile, France) with a confidence value of 99% according to the manufacturer’s instruction. For the MALDI-TOF MS analysis, the samples tested were confirmed as *S. canis* with commercial databases, with a log(score) > 2.0, which indicates that identification is reliable at species level. The dilution susceptibility test was performed for the three pure strains, and they were susceptible to all the antimicrobials tested except for the tetracycline. All the samples tested with the Real-time PCR were negative for the above-mentioned mastitis-related pathogens.

### 3.2. Follow-Up

Positively tested animals were treated with antibiotic Synulox Lactating Cow (amoxicillin trihydrate 200 mg, potassium clavulanate 50 mg, prednisolone 10 mg), and Zoetis Italia S.r.l. (Catania, Italy), following the manufacturer’s instructions. Antibiotic therapy was administered in accordance with the prescriptions provided by the manufacturer, after consulting the veterinarian treating farm. The antibiotic was administered, by intramammary injection every 12 h, for three consecutive milking sessions. Farmers were briefed on proper hygiene practices to be applied on the farm to improve milking routines (e.g., use of pre-dipping and post-dipping and implementation of biosecurity measures—no free access for the domestic carnivores present on the farm). After first pathogen isolation and antibiotic treatment for positive animals (~30 days), the milk collection was repeated for these animals (3 samples) adding all remaining lactating cows present in the herd (23 samples). A total of 26 cows were then subsequently tested, with the purpose of evaluating the treatment effectiveness and the health status of the untreated animals.

### 3.3. Second Isolation

The CMT was repeated for all the 26 milk samples, with 9 resulting positive. The SCC was also evaluated. The SCC was >200 × 10^3^ cell/mL in three cows, respectively, sample 4 (552 × 10^3^ cell/mL), sample 13 (2723 × 10^3^ cell/mL), and sample 21 (998 × 10^3^ cell/mL), confirming the presence of subclinical mastitis. The same procedures of bacteria isolation and characterization were adopted for all the samples: pure colonies of GGS were obtained from the three samples with high SCC. The API Strep system (bioMerieux, Marcy-l’Étoile, France) and MALDI-TOF MS methodology confirmed the presence of *S. canis* in the three samples. Sample n. 4 was tested also the first isolation confirming the persistence of mastitis in this cow. The dilution susceptibility test was performed for the three pure strains having the same pattern of resistance coming from the first isolation. The same results of the Real-time PCR were detected for all 26 samples. In the herd examined in the present study, 5 animals out of the 31 lactating cows tested in the first and second sampling were infected, with mastitis having a prevalence of 16.12% (95% C.I. 5.5% to 33.7%). Considering the CMT positive samples, three out of eight animals (first sampling) and three out of nine animals (second sampling) were infected. After the second isolation, the farmyard dogs were clinically examined and swabs from the nasal or pharyngeal mucosa were collected for evaluating the possible role as a contaminant and cultured onto Columbia blood agar (CBA), Mannitol salt agar (MSA), MacConkey agar (MCK), and Modified Edwards medium plates (ED) (Liofilchem, Teramo, Italy) and incubated at 37 °C for 24–48 h in aerobic conditions. No colony reportable to *S. canis* was found. Sample collection from circulating cats was unsuccessful as they did not easily approach humans.

### 3.4. Biomolecular Analysis

Genomic DNA isolated from four single colonies of *S. canis* (samples 2, 4, and 7 coming from the first isolation, T0, and sample 21 from the second isolation, T1, Table 1) were extracted using a MagMAX CORE Nucleic Acid Purification Kit, according to the manufacturer’s protocol. The DNA concentrations were estimated by a Qubit Fluorometer using Qubit dsDNA HS Assay (Thermo Fisher Scientific, Waltam MA, USA). The paired-end genomic libraries were prepared using an Illumina DNA Prep Kit (Illumina, San Diego, CA, USA). Sequencing was performed using a MiSeq Reagent Kit v2 (2 × 250 bp) on an Illumina MiSeq platform (Illumina, San Diego, CA, USA) as described by Farina et al. [18]. Genomic sequence characterization of the isolated samples was performed through a customized Galaxy pipeline, consisting of quality control (FastQC [22], fastp v0.20.1 [23]); de novo genome assembly (UniCycler v.0.5 [20]) and quality check (QUAST v.5.2 [24]); contaminant sequence search (CheckM v1.2.0 [25]); and sequencing coverage analysis (Bowtie2 aligner v2.5.0 [26], Qualimap v.2.2.2 [27]).

### 3.5. Genotyping and Genomic Characterization

MLST allele profiles for all PubMLST *S. canis* isolates were given as input for integrated GrapeTree and BURST analysis tools (https://pubmlst.org/bigsdb?db = pubmlst_scanis_isolates, accessed on 3 March 2025), in order to evidence genetic relationships, host specificity, and geographical distribution of the isolates with respect to the case study.

Figure 1 summarizes the available genetic information for this species. *S. canis* isolated from cattle (sample 21) belongs to ST24, clustering with five samples isolated from dogs. ST24 cluster differs for one allele from the ST1, ST53, ST50, and ST70 groups, mainly isolated from dogs among Europe and Japan. It differs by more than one allele from the twelve Pub-MLST recorded Italian samples. The MLST allele profile and metadata for sample 21 genome have been deposited into PubMLST (submission date: 24 October 2024).

The assemblies were confirmed to belong to *S. canis* (>97% ANIBlast similarity to *Streptococcus canis* NCTC12191 strain genome) and were highly similar to each other (>99.9% ANIBlast pairwise similarity across the four isolates). Two antibiotic-resistant genes, *lmrP* and *TetM*, were predicted with >99% similarity to reference sequences, according to antibiotic-resistant gene prediction analysis.

## 4. Discussion

In this case study, we report the isolation and the phenotypic and genotypic characterization of *S. canis* originating from subclinical mastitis in a herd. In the dairy production industry, udder health problems can cause of economic losses. *S. canis* is morphologically and epidemiologically similar to *S. agalactiae* [12]. For this study, colonies were identified using MALDI-TOF MS technology, a reliable method for identifying *S. canis* and other β-hemolytic streptococci. However, accurate species identification requires a comprehensive and up-to-date database, as highlighted by Nybakken et al. (2021) [28]. Implementation of mastitis control programs in many countries has reduced infections of the bovine udder caused by main pathogens such as *S. agalactiae*. IMI caused by GGS is rare, with a reported incidence of isolation of <1% [12]. Outbreaks are reported from all over the world but the origin of such as an infection is still not clear [11,29,30]. Information regarding the epidemiological route of entry of *S. canis* into the bovine host, the course of the disease within the herd, and cow-to-cow transmission is scarce. It is speculated that infection may result from transmission from other animals, such as dogs and cats, and subsequently spread from cow to cow in contagious manner as a result of poor milking hygiene [8,11,29,30,31]. The data reported are in line with publications already present in the literature and would lead us to consider *S. canis* as a contagious pathogen causing subclinical mastitis in dairy cattle [29]. The genetic studies carried out on the strains isolated in the two samplings in our study confirm this evidence. In particular, the two strains are identical and were identified as belonging to the sequence type ST24. This finding confirms the circulation of the bacterium on the farm even after the implementation of management procedures to prevent the spread of the disease and after the therapy of the first detected animals. Circulation of the bacterium even after antibiotic treatment of positive animals and improved hygienic conditions may be attributable to several hypotheses. Transmission is related to the presence of dogs and cats on farms, being a possible persistent source infection [13]. ST24 appears to be reported only in dog samples, and this might be in line with that hypothesis, as recorded in the PubMLST data bank. A combined analysis of several studies suggests that the genetic diversity of *S. canis* plays a crucial role in understanding its epidemiology, pathogenicity, and zoonotic potential [32,33,34]. Among these, a new isolate of *S. canis* ST55 has been associated with a case of subclinical mastitis in a dairy herd, which poses a potential emerging risk in subclinical mastitis infections [35]. In our study, farmyard dogs were tested, while this was not possible for cats. It is worth noting that *S. canis* can occur within farms without the presence of carnivores reported in the barn [27], indicating possible indirect introduction via the farmer’s skin (or that of a different person) working in the barn. Whereas knowledge about the *S. canis* epidemiology in human medicine is based on a small number of studies, the zoonotic potential is now widely accepted. In fact, human infections are reported to be rare, although there has recently been an increase in reported cases [36,37,38,39]. This study also reports that the microorganism was still detected in 2 cows in a total of 26 infected cows after a 6-month resampling. The data provided are not clear about the possible reinfection of the two cows or the refractory infection to the previous treatment. A study reports that *S. canis* is able to spread from cow to cow in a contagious manner [8]; after the first detection of *S. canis*, dogs were removed from the barn; in addition, the farmer built fences to prevent animals other than cattle species from entering the barn. We can therefore assume that the chronically infected cows were the only possible source for the animals in the second and third survey. This was supported by the result of strain genotyping. One of the infected cows was not responsive to two series of antimicrobial treatments, suggesting a possible long-term infection and emphasizing the importance of detection of these carriers by re-examination. Although our study does not identify the infection source, it can confirm the hypothesis of long-term infection in a cow with a failed treatment that was responsible for the new infections in the barn. Also, the microorganism was found to be susceptible to all the antimicrobials tested in vitro except for tetracycline. The treatment could be ineffective and the cure rate in infected quarters could be 89% [28], with dry treatment facilitating better cure rates [13]. Further studies should be carried out to increase knowledge about *S. canis* epidemiology, considering its potential contagiousness and strong similarity with *S. agalactiae,* in terms of high infectivity and low self-cure rate [10,13,40].

## 5. Conclusions

This paper reports a clinical case of subclinical mastitis caused by *S. canis* with increased SCC in a dairy herd. The diagnosis was possible with conventional bacteriology and genotyping using MLST. Treatment of the first positive cows without the adoption of the optimal management and hygiene measure was not effective. The source of infection was not identified but cow-to-cow transmission and a long-term infection were detected. For the successful eradication of *S. canis* within the herd, straight management and control instructions are needed, like those applied in *S. agalactiae* programs. Early detection and immediate treatment of infected cows are advised and improvement of removal of infected long-term shedding in animals, milking hygiene, and management is necessary to avoid new or reinfections. Despite some limitations, this work adds an important contribution to the literature on subclinical mastitis related to *S. canis* in dairy cattle. Further studies should be performed for a better understanding of the epidemiology and availability of more accurate data about *S. canis* infections in dairy cows.

## Figures and Tables

**Figure 1 vetsci-12-00286-f001:**
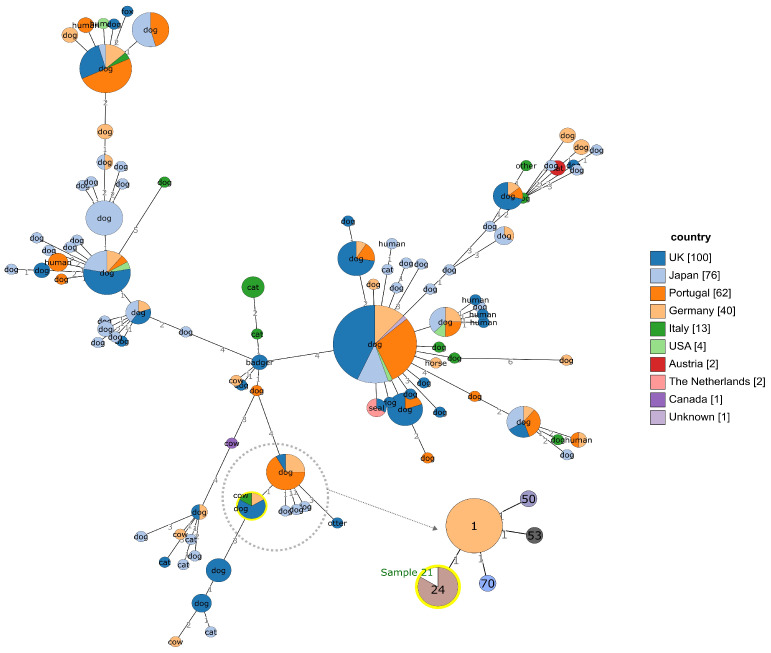
Minimum spanning tree for *S. canis* isolates, as generated by PubMLST GrapeTree and BURST tools. Main tree: nodes are colored according to geographical origin of the isolate; common name of host species within each node; integer numbers on branches: number of different 7-gene MLST alleles between nodes. Tree extract on the lower right side: nodes with 1-allele difference with the ST24 node, in which isolate from sample 21 is placed. Relevant node is evidenced in yellow.

**Table 1 vetsci-12-00286-t001:** Results from diagnostic tests performed on 31 milk samples in the first (T0) and second (T1) sampling. Californian Mastitis Test (CMT), somatic cell count (SCC), isolation (ISOL.), whole-genome sequencing (WGS), negative (N), positive (P), not tested (NT).

	T0			T1			
	CMT	SCC	ISOL.	CMT	SCC	ISOL.	WGS
Sample 1, 3, 5, 6, 8	P	N	N	NT	NT	NT	NT
Sample 2, 7	P	P	P	N	N	N	T0
Sample 4	P	P	P	P	P	P	T0
Sample 10, 16, 17, 24, 25, 28	N	NT	NT	P	N	N	NT
Sample 13	N	NT	NT	P	P	P	NT
Sample 21	N	NT	NT	P	P	P	T1
Sample 9, 11, 12, 14, 15, 18, 19, 20, 22, 23, 26, 27, 29, 30, 31	N	NT	NT	N	N	N	NT

## Data Availability

MLST allele sequences and metadata for sample 21 are deposited in PubMLST, (https://pubmlst.org/bigsdb?db=pubmlst_scanis_isolates) (accessed on 24 October 2024), Identifier: 301. MLST allele sequences and metadata of all sequenced samples are deposited in PubMLST, (https://pubmlst.org/bigsdb?db=pubmlst_scanis_isolates) (accessed on 3 March 2025). Genome assembly sequences for sample 2, 4, 7, and 21 have been uploaded to NCBI, BioProject accession: PRJNA1231738.

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
