# Peer review of "Subclinical Mastitis Related to Streptococcus canis Infection in Dairy Cattle"

_vetsci, 2025, doi:10.3390/vetsci12030286_

Round 1

Reviewer 1 Report

Comments and Suggestions for Authors

I have read the manuscript with interest. I feel that such a thorough investigation of environmental pathogens is rarely done in mastitis diagnosis. Therefore, Streptococcus canis infections may go unnoticed. I would further investigate the cause of nonresponse to long antibiotic treatments, which is now mentioned as only a side note in the manuscript.

I find the overall quality of the manuscript to be very good. The paper can, however, make use of a few corrections in grammar and structure.

I listed my suggestions below.

Simple summary:
I would remove the second sentence (Causes include physical, chemical...) I think it conveys a false idea of mastitis to lay readers.

Abstract: After revision of the main text, I suggest to revise the abstract to be a tiny bit more story-telling. Now it is a mere enumeration of short concise sentences which could be made more coherent parts of a whole. But this is only a suggestion.

Some sentences do need some grammatical touch-ups, though: 
L35-36: "the presence of two other cows were positive"

L39: The gene ..... was predicted.

Introduction:

A grammatical revision is needed for :

L55: "These microorganisms are also founded in other animals" 

L58: it can represents

(Such errors can be found here and there throughout the text which necessitates a thorough language check-up. I will not list all specific sentences later on.)

I suggest writing out "mass spectrometry" after MALDI-TOF at the end of the Intro. The abbreviation "MS" comes up in later parts of the text but is not written out.

Materials and Methods:

It would be useful to know the milk yield of cows and how much it decreased, which suggested mastitis to the farmer.
With such apparent environmental risks, were there no history of mastitis on the farm and potentially its causes? Now it seems that this suspicion of subclinical mastitis just appeared out of the blue, which is interesting given the low level of mastitis awareness. You may write a sentence or two about the "anamnesis", if you agree.

Please indicate why you didn't wish to sample the milking equipment for the presence of pathogens.

Also, please be a bit more detailed on further visits. Results say Second isolation which is not apparent from the Mat and Meth section.

Results:

You may supplement the results on the remaining 5 animals that tested positive for CMT. Were they not examined further due to low SCC? Please clarify.

Please specify the dose, frequency and way of administration of Synulox. Given that it was not efficient in certain cases these details would be informative.

Do you have any information on the effect of the education of the farmer? The manuscript suggests that follow-up visits were made to the farm. Was there any improvement in milking hygiene?

Results - follow-up: It sounds more like Methods in some parts. consider moving some parts to Mat and Meth. (see my last comment to M&M)

Lines 194-196: “Were infected” and prevalences are referred to multiple times, but it is not clarified what they refer to exactly or why they are necessary. 

Discussion

Line 271: dogs were removed from the barn. Were cats also removed?

Line 275: The futility of a second cycle of AB treatment is an important result. I suppose it could, therefore, be mentioned in the Results. What antibiotic was used for the second time? (I'm sorry if I missed it if was there somewhere.) In the discussion, it is worth exploring this a little bit more, and this is why I indicated writing more details about the antibiotic therapy.

Else: do authors wish to compare the calculated prevalences reported in Lines 194-196 to other studies? Without it, reporting prevalences seems a bit pointless.

Line 282: Tikofsky and Zadoks mention the ineffectiveness of AB treatment. Were the reported causes present on this farm? What could cause the lack of response of an otherwise susceptible pathogen?

Line 284: Authors mention the strong similarity to S. agalactiae. Is crossreaction in diagnostic methods possible?

The references mentioning reports on Streptococcus canis are not so recent. Although this publication below is in Hungarian, the abstract is in English. It is a more recent isolation of S. canis in mastitis.

https://www.webofscience.com/wos/woscc/full-record/000372672700002

Authors might make use of it.

Thank you for taking my points into consideration.

Comments on the Quality of English Language

I have made some remarks on the grammar in the general review part

Author Response

I have read the manuscript with interest. I feel that such a thorough investigation of environmental pathogens is rarely done in mastitis diagnosis. Therefore, Streptococcus canis infections may go unnoticed. I would further investigate the cause of nonresponse to long antibiotic treatments, which is now mentioned as only a side note in the manuscript. I find the overall quality of the manuscript to be very good. The paper can, however, make use of a few corrections in grammar and structure.

I listed my suggestions below.

Simple summary:
Comment 1: I would remove the second sentence (Causes include physical, chemical...) I think it conveys a false idea of mastitis to lay readers.

Response 1: Amended. We removed the entire sentence.

Abstract:

Comment 2: After revision of the main text, I suggest to revise the abstract to be a tiny bit more story-telling. Now it is a mere enumeration of short concise sentences which could be made more coherent parts of a whole. But this is only a suggestion.

Response 2: Thank you for the suggestion. We reviewed the abstract ad merged some sentences in lines 29-35, 37-43, 48-49.

Comment 3: Some sentences do need some grammatical touch-ups, though: 
L35-36: "the presence of two other cows were positive"

L39: The gene ..... was predicted.

Response 3: Amended. We modified these sentences in lines 41-43, 47.

Introduction:

Comment 4: A grammatical revision is needed for :

L55: "These microorganisms are also founded in other animals" 

L58: it can represents

Response 4: Amended. We modified the text in lines 69-70, 73.

(Such errors can be found here and there throughout the text which necessitates a thorough language check-up. I will not list all specific sentences later on.)

Comment 5: I suggest writing out "mass spectrometry" after MALDI-TOF at the end of the Intro. The abbreviation "MS" comes up in later parts of the text but is not written out.

Response 5: We removed the sentence in the introduction but added “mass spectrometry” in “materials and methods,” lines 147-148.

Materials and Methods:

Comment 6: It would be useful to know the milk yield of cows and how much it decreased, which suggested mastitis to the farmer.
With such apparent environmental risks, were there no history of mastitis on the farm and potentially its causes? Now it seems that this suspicion of subclinical mastitis just appeared out of the blue, which is interesting given the low level of mastitis awareness. You may write a sentence or two about the "anamnesis", if you agree.

Response 6: Thank you for the suggestion. We asked to the farmer some information about cattle milk production. We added proper sentences in lines 95-97. Furtermore, cases of subclinical mastitis had already been reported on the farm, but no sampling had been carried out. After our analysis, we detected the presence of S. canis, as mentioned in the manuscript (lines 119-121).

Comment 7: Please indicate why you didn't wish to sample the milking equipment for the presence of pathogens.

Response 7: Added sentence: sampling was done exclusively from the animals in order to ensure sterile conditions during the sampling procedure. Lines 118-119. Furthermore, milking equipment was not tested due to lack of availability from the farmer.

Comment 8: Also, please be a bit more detailed on further visits. Results say Second isolation which is not apparent from the Mat and Meth section.

Response 8: We added more information in materials and methods (lines 119-121)

Results:

Comment 9: You may supplement the results on the remaining 5 animals that tested positive for CMT. Were they not examined further due to low SCC? Please clarify.

Response 9: During the first isolation, only animals suspected of infection were tested. Specifically, bacteriological examination was performed on all eight samples 8 animals and 3 of them tested positive for S.canis. The other five samples were proved to be contamination-free with mastidogenic agents, as they were found to be sterile on bacteriological examination.

Comment 10: Please specify the dose, frequency and way of administration of Synulox. Given that it was not efficient in certain cases these details would be informative.

Response 10: We added some information about the administration of the antibiotic in the text. Lines 213-216.

Comment 11: Do you have any information on the effect of the education of the farmer? The manuscript suggests that follow-up visits were made to the farm. Was there any improvement in milking hygiene?

Response 11: Good hygiene practices were not implemented, in fact during the second sampling we found new positive animals.

Comment 12: Results - follow-up: It sounds more like Methods in some parts. consider moving some parts to Mat and Meth. (see my last comment to M&M)

Response 12: We moved some parts of results in matherial and methods. From lines 266-274 to 178-185.

Comment 13: Lines 194-196: “Were infected” and prevalences are referred to multiple times, but it is not clarified what they refer to exactly or why they are necessary. 

Response 13: Thanks for the comment. We removed the percentages.

Discussion

Comment 14: Line 271: dogs were removed from the barn. Were cats also removed? 

Response 14: The entry of cats was prevented by fencing off the barn. We added some information in lines 352-353.

Comment 15: Line 275: The futility of a second cycle of AB treatment is an important result. I suppose it could, therefore, be mentioned in the Results. What antibiotic was used for the second time? (I'm sorry if I missed it if was there somewhere.) In the discussion, it is worth exploring this a little bit more, and this is why I indicated writing more details about the antibiotic therapy.

Response 15:  Treatment was carried out only one time. After treatment we found one cow with chronic infection and two new positive.

Comment 16: Else: do authors wish to compare the calculated prevalences reported in Lines 194-196 to other studies? Without it, reporting prevalences seems a bit pointless.

Response 16:  We removed the prevalence regarding the CMT result, but we tested it for indications of somatic cell count

Comment 17: Line 282: Tikofsky and Zadoks mention the ineffectiveness of AB treatment. Were the reported causes present on this farm? What could cause the lack of response of an otherwise susceptible pathogen?

Response 17: Tikofsky and Zadoks reported that infected animals are found to be susceptible to all antibiotics tested. This study was mentioned in our manuscript to indicate the best productive statement to carry out antibiotic treatment in order to increase cure rates.

Comment 18: Line 284: Authors mention the strong similarity to S. agalactiae. Is crossreaction in diagnostic methods possible?

Response 18: The similarity to S. agalactiae was mentioned exclusively in terms of its contagious potential and similar clinical signs.

Comment 19: The references mentioning reports on Streptococcus canis are not so recent. Although this publication below is in Hungarian, the abstract is in English. It is a more recent isolation of S. canis in mastitis.

https://www.webofscience.com/wos/woscc/full-record/000372672700002

Response 19: Thank you for the suggestion. We added the new reference in lines 322.

Thank you for your interesting advising and considerations.

Reviewer 2 Report

Comments and Suggestions for Authors

Comments of the Reviewer on the manuscript vetsci-3468437

The present study presents a case report about an outbreak of subclinical mastitis in a dairy herd caused by Stretococcus (S.) canis. S. canis isolates were characterized using the latest molecular biological methods.

Presentations of S. canis infections make an important contribution to animal and human health due to the wide host range, the great impact on infected dairy herds and he zoonotic potential of this pathogen.

This present case report is a comprehensive and concise presentation of this outbreak and a valuable contribution to bovine mastitis caused by S. canis.

There are some shortcomings that should be addressed.

General comments

Linguistic revision by a native speaker is recommended.

A more in depth evaluation of the elaborated data and presentation of the results would enhance the significance of this valuable study.

Whole genome sequences are not provided on the platform PubMLST. However, the whole genome sequences of the samples 2, 4, 7, and 21 should be provided in the NCBI genome database. Whole genome sequences are essential for molecular epidemiological studies and represent an important enrichment of the sequence database.

A Minimum Spanning Tree (MST) (e.g. goeBURST diagram or cgMLST) including whole genome sequence data from S. canis isolated from different host species like, dairy cows, dogs, cats, and humans would illustrate the results clearly (see Fukushima Y, Takahashi T, Goto M, Yoshida H, Tsuyuki Y. Novel diverse sequences of the Streptococcus canis M-like protein (SCM) gene and their prevalence in diseased companion animals: Association of their alleles with sequence types. J Infect Chemother. 2020 Sep;26(9):908-915. doi: 10.1016/j.jiac.2020.04.004. Epub 2020 Apr 28. PMID: 32354600; Pinho MD, Foster G, Pomba C, Machado MP, Baily JL, Kuiken T, Melo-Cristino J, Ramirez M; Portuguese Group for the Study of Streptococcal Infections. Streptococcus canis are a single population infecting multiple animal hosts despite the diversity of the universally present M-Like protein SCM. Front Microbiol. 2019 Mar 29;10:631. doi: 10.3389/fmicb.2019.00631. PMID: 30984150; PMCID: PMC6450190)

Furthermore, a minimum spanning tree (MST) based on cgMLST data would characterize S. canis in an unprecedented depth for the first time to the best of my knowledge.

Replace “genotypization” with the with the more common term “genotyping”.

Special comments

Chapter Simple Summary

Line 16: in order to avoid repetitions replace “Infectious agents” with “Pathogens”.

Line 17: start the sentence with “Among these, Streptococcus canis”.

Lines 25-27: remove “and the possibility that animals can be a source of infection for humans” due to repetition.

Line 29: replace “on CMT” with “in California Mastitis Test (CMT)”.

Line: 30: add “the” before “causative agent”, add “In addition,” before “the SCC”, and replace “SCC” with “somatic cell count (SCC)”.

Line 33: what is the meaning of “strains were extracted”? Please specify.

Line 34: replace “at the qPCR” with “in qCR”.

Line 35: add “MS” behind “MALDI-TOF”.

Line 35-36: rephrase the sentence “The resistant …” with “In susceptibility testing resistance against tetracycline was detected”.

Lines 36-37: This sentence is difficult to understand. Please rephrase.

Keywords

Remove the keyword “diagnosis” and add “whole genome sequencing”.

Chapter Introduction

At least 40 MLST sequence types have been described for S. canis. Please consider this information in the chapter Introduction and Discussion(see Fukushima Y, Takahashi T, Goto M, Yoshida H, Tsuyuki Y. Novel diverse sequences of the Streptococcus canis M-like protein (SCM) gene and their prevalence in diseased companion animals: Association of their alleles with sequence types. J Infect Chemother. 2020 Sep;26(9):908-915. doi: 10.1016/j.jiac.2020.04.004. Epub 2020 Apr 28. PMID: 32354600)

Lines 47-48: rephrase this sentence for better understanding and replace “Camp” with “CAMP”.

Lines 53-54: Replace this sentence with “The zoonotic pathogen has been isolated also in human genital, urinary and upper respiratory tract infections through transmission by dogs [5], [6]”.

Line 58: replace “it can represents” with “S. canis can represent”.

Line 68: add a space between “level” and “[8]”.

Lines: 71-74: cow-to-cow transmission cannot be detected or verified by the methods listed, except for molecular typing. Please specify.

Lines 83-86: The meaning of the two sentences is unclear. Please rephrase.

Chapter Material and Methods

Resampling after 6 months is not mentioned in this chapter (s. subchapter “Follow‐up” and line 268 in the chapter Discussion).

Line 78: replace “for a possible” with “for cases of possible”.

Lines 83-86: please rephrase for better understanding and readability.

Line 93: replace “California Mastitis Test (CMT)” with “CMT” (s. comment on line 29).

Line 96: replace “on CMT” “in the CMT”.

Line 97: add “the” before “National Mastitis Council”.

Line 105: replace “with aerobic conditions” with “under aerobic conditions”.

Lines 105-107: reliable visual identification of bacteria at species level is not possible. Therefore, rephrase this sentence as follows: Bacterial cultures were inspected visually and growth of one or more uniform colonies were considered relevant and growth of three and more different colonies were considered as contamination.

Lines 109-110: remove the sentence “Catalase test was evaluated” and insert “catalase test” behind “Gram stain”.

Line 110: remove “serological”

Line 112: replaces “doupt” with “doubtful” or “questionable”.

Line 114-119 (subchapter “MALDI‐TOF MS methodology”). Several β-haemolytic streptococcal species are difficult to identify properly (Nybakken EJ, Oppegaard O, Gilhuus M, Jensen CS, Mylvaganam H. Identification of Streptococcus dysgalactiae using matrix-assisted laser desorption/ionization-time of flight mass spectrometry; refining the database for improved identification. Diagn Microbiol Infect Dis. 2021 Jan;99(1):115207. doi: 10.1016/j.diagmicrobio.2020.115207. Epub 2020 Sep 22. PMID: 33069003). Therefore the specification of the Bruker database used is relevant. Did you use the the improved version Brukers Compass Library DB-7854?

Lines 115-116: reword this sentence as follows: Bacterial colonies tested with ambiguous results were subjected to species identification using MALDI-TOF MS. Please consider this issue also in the chapter “Discussion”.

Chapter Results

Data for testing of the dogs is missing (s. lines 260-261).

Lines 205-219: the subchapter “Biomolecular analysis” describes technical details and should be moved to the chapter “2. Materials and Methods”

Line 208: Table 1 should be included in the manuscript text with a meaningful legend.

Lines 221-229: The subchapter „Genotypization and genomic characterization“ contains technical details and should be moved to the chapter “2. Materials and Methods”.

Chapter Discussion

The isolation of different MLST sequence types in dairy cows, dogs, cats and humans should be discussed in more detail. Thus, please consider the occurrence of the sequence type ST55 in a dairy herd suffering from subclinical mastitis (Eibl C, Baumgartner M, Urbantke V, Sigmund M, Lichtmannsperger K, Wittek T, Spergser J. An Outbreak of Subclinical Mastitis in a Dairy Herd Caused by a Novel Streptococcus canis Sequence Type (ST55). Animals (Basel). 2021 Feb 20;11(2):550. doi: 10.3390/ani11020550. PMID: 33672442; PMCID: PMC7923261; Richards VP, Zadoks RN, Pavinski Bitar PD, Lefébure T, Lang P, Werner B, Tikofsky L, Moroni P, Stanhope MJ. Genome characterization and population genetic structure of the zoonotic pathogen, Streptococcus canis. BMC Microbiol. 2012 Dec 18;12:293. doi: 10.1186/1471-2180-12-293. PMID: 23244770; PMCID: PMC3541175; Pagnossin D, Weir W, Smith A, Fuentes M, Coelho J, Oravcova K. Streptococcus canis genomic epidemiology reveals the potential for zoonotic transfer. Microb Genom. 2023 Mar;9(3):mgen000974. doi: 10.1099/mgen.0.000974. PMID: 37000493; PMCID: PMC10132062.)

Line 239: replace “coming from” with “originating from”.

Line 240: replace “could be cause” by “can cause”, delete “of” and add “a” before “contagious”.

Lines 241: Please insert a comment on identification of S. canis using MALDI-TOF MS: MALDI-TOF MS is a reliable method for identification and differentiation of β-haemolytic streptococcal species. However, for precise species identification of those bacteria a comprehensive data base is required

(Nybakken et al. 2021).

Lines 118-119: replace “compared with those of known microbial isolates of the commercial library provided by Bruker Daltonics” with “compared with those of the commercial database provided by Bruker Daltonics (Milano, Italia)”.

Line 138-142: put the additional information “‐including all major coagulase‐negative staphylococci‐”, “-including E. faecalis and E. faecium-” and “‐penicillin‐resistance gene‐” in brackets.

Line 143: remove the bracket behind “Prototheca spp.)”.

Line 249-251: the meaning of this sentence is unclear. Please rephrase for better understanding.

Line 250: delete the space after “[28)”.

Line 252: remove “can” before “confirm”. Finish the sentence after the word “evidence” and start the next sentence with “In particular”.

Line 255: replace “therapy in” with “therapy of”.

Line 258: replace “being the possible source” with “being a possible persistent source”.

Line 260: put “the” before “PubMLST”.

Line 261: replace “while cats were impossible to evaluate” with “while this was not possible for cats”.

Line 266: replace “understood” with “reported”.

Line 267: insert a space between “cases” and “[30]”. Replace “The study” with “This study” or “Our study”.

Line 274: replace “cows infected” with “infected cows”.

Line 275: replace “the possible” with “a possible” and insert “emphasizing” before “the importance”.

Line 276: replace “with a re-examination” with “by re-examination. Remove “of the treated cows for the control of the infection”.

Line 279: replace “antimicrobial” with “antimicrobials”.

Lines 279-283: this sentence is difficult to understand. Please rephrase in shorter sentences.

Line 289: replace “Multi Sequence Typing” with “genotyping using MLST”.

Line 294: replace “the improve” with “improvement of” and insert the recommendation “removal of infected long-term shedding in animals”.

Line 295: replace “re-infection” with “re-infections”.

Line 297: replace “to better understand” with “for better understanding”.

Line 298: remove “and to have more accurate about that infections” with “and availability of more accurate data about S. canis infections”.

Comments on the Quality of English Language

Linguistic revision by a native speaker is recommended.

Author Response

The present study presents a case report about an outbreak of subclinical mastitis in a dairy herd caused by Stretococcus (S.) canisS. canis isolates were characterized using the latest molecular biological methods.

Presentations of S. canis infections make an important contribution to animal and human health due to the wide host range, the great impact on infected dairy herds and he zoonotic potential of this pathogen.

This present case report is a comprehensive and concise presentation of this outbreak and a valuable contribution to bovine mastitis caused by S. canis

There are some shortcomings that should be addressed.

General comments

Comment 20: Linguistic revision by a native speaker is recommended.

Response 20: We made some revisions to the paper and modified it according to the authors' suggestions

Comment 21: A more in depth evaluation of the elaborated data and presentation of the results would enhance the significance of this valuable study.

Whole genome sequences are not provided on the platform PubMLST. However, the whole genome sequences of the samples 2, 4, 7, and 21 should be provided in the NCBI genome database. Whole genome sequences are essential for molecular epidemiological studies and represent an important enrichment of the sequence database.

Response 21: Thanks for the suggestion. We have submitted the draft genome sequence of S. canis isolate (Sample 21 within the manuscript, PubMLST identifier 301). Assembly sequences for the remaining samples are basically the same isolate, with 99.99% Average Nucleotide Identity with the Sample 21. Thus, we did not submit them to NCBI resource. Draft genome assembly for this sample can be found at https://www.ncbi.nlm.nih.gov/bioproject/PRJNA1231738 (FASTA file will be public upon NCBI sequence processing).

Comment 22: A Minimum Spanning Tree (MST) (e.g. goeBURST diagram or cgMLST) including whole genome sequence data from S. canis isolated from different host species like, dairy cows, dogs, cats, and humans would illustrate the results clearly (see Fukushima Y, Takahashi T, Goto M, Yoshida H, Tsuyuki Y. Novel diverse sequences of the Streptococcus canis M-like protein (SCM) gene and their prevalence in diseased companion animals: Association of their alleles with sequence types. J Infect Chemother. 2020 Sep;26(9):908-915. doi: 10.1016/j.jiac.2020.04.004. Epub 2020 Apr 28. PMID: 32354600; Pinho MD, Foster G, Pomba C, Machado MP, Baily JL, Kuiken T, Melo-Cristino J, Ramirez M; Portuguese Group for the Study of Streptococcal Infections. Streptococcus canis are a single population infecting multiple animal hosts despite the diversity of the universally present M-Like protein SCM. Front Microbiol. 2019 Mar 29;10:631. doi: 10.3389/fmicb.2019.00631. PMID: 30984150; PMCID: PMC6450190)

Response 22: Thank you for your valuable suggestion. We have scanned PubMLST, retrieving 301 S. canis isolate (including the one described in this paper) and produced a summarizing Grape Tree image, with isolates clustered according their MLST Sequence Type, coloured by country of origin. Numbers in branch represent allelic difference between nodes (ST). Sequence Types with one-allele difference has been obtained by using BURST plugin in PubMLST resource. Details together with comments are present from line 274 to 290.

Comment 23: Furthermore, a minimum spanning tree (MST) based on cgMLST data would characterize S. canis in an unprecedented depth for the first time to the best of my knowledge. 

Response 23: Thanks for this advice. We searched public databases for S. canis genomes: to date, there are only around 50 public genome assemblies (NCBI). According to our expertise and pertinent literature, a reliable cgMLST scheme should be based on hundreds of genome assemblies. The ChewBBACA suite (https://chewbbaca.readthedocs.io/en/latest/user/tutorials/chewie_step_by_step.html) requires good quality genomes in order to detect a consistent core gene set and identify alleles. Nonetheless, we will constantly monitor public resources and we will try to generate a preliminary cgMLST scheme, in the event that 80-100 genomes will be globally present.

Comment 24: Replace “genotypization” with the with the more common term “genotyping”.

Response 24: Amended. We changed “genotypization” in “genotyping”, line 266.

Special comments

Chapter Simple Summary

Comment 25: Line 16: in order to avoid repetitions replace “Infectious agents” with “Pathogens”.

Response 25: Amended, line 16.

Comment 26: Line 17: start the sentence with “Among these, Streptococcus canis”.

Response 26: Amended, line 17.

Comment 27: Lines 25-27: remove “and the possibility that animals can be a source of infection for humans” due to repetition.

Response 27: we removed this sentence.

Comment 28: Line 29: replace “on CMT” with “in California Mastitis Test (CMT)”.

Response 28: Amended, line 29.

Comment 29: Line: 30: add “the” before “causative agent”, add “In addition,” before “the SCC”, and replace “SCC” with “somatic cell count (SCC)”.

Response 29: Based on the first reviewer's comments, we modified this sentence, line 30.

Comment 30: Line 33: what is the meaning of “strains were extracted”? Please specify.

Response 30: We specified that the DNA of the bacterial strains was extracted. Line 25.

Comment 31: Line 34: replace “at the qPCR” with “in qCR”.

Response 31: We moved and edited the sentence. Line 33.

Comment 32: Line 35: add “MS” behind “MALDI-TOF”.

Response 32: Amended, line 39.

Comment 33: Line 35-36: rephrase the sentence “The resistant …” with “In susceptibility testing resistance against tetracycline was detected”.

Response 33: Amended, lines 41-43.

Comment 34: Lines 36-37: This sentence is difficult to understand. Please rephrase.

Response 34: We modified the entire sentence, lines 41-43.

Keywords

Comment 35: Remove the keyword “diagnosis” and add “whole genome sequencing”.

Response 35: Amended, line 53.

Chapter Introduction

Comment 36: At least 40 MLST sequence types have been described for S. canis. Please consider this information in the chapter Introduction and Discussion (see Fukushima Y, Takahashi T, Goto M, Yoshida H, Tsuyuki Y. Novel diverse sequences of the Streptococcus canis M-like protein (SCM) gene and their prevalence in diseased companion animals: Association of their alleles with sequence types. J Infect Chemother. 2020 Sep;26(9):908-915. doi: 10.1016/j.jiac.2020.04.004. Epub 2020 Apr 28. PMID: 32354600)

Response 36: Thank you for the suggestion. We have included the reference in the discussion (line 338). Additionally, the samples analyzed in this study (Fukushima et al.) were considered for constructing the phylogenetic tree (Minimum Spanning Tree in line 285), as they are publicly available samples from PubMLST.

Comment 37: Lines 47-48: rephrase this sentence for better understanding and replace “Camp” with “CAMP”.

Response 37: Amended, lines 58-59.

Comment 38: Lines 53-54: Replace this sentence with “The zoonotic pathogen has been isolated also in human genital, urinary and upper respiratory tract infections through transmission by dogs [5], [6]”.

Response 38: Amended, lines 65-67.

Comment 39: Line 58: replace “it can represents” with “S. canis can represent”.

Response 39: Amended, line 72-73.

Comment 40: Line 68: add a space between “level” and “[8]”.

Response 40: Amended, line 83.

Comment 41: Lines: 71-74: cow-to-cow transmission cannot be detected or verified by the methods listed, except for molecular typing. Please specify.

Response 41: Thank for the suggestion. We delete other methods. Lines 88-90.

Comment 42: Lines 83-86: The meaning of the two sentences is unclear. Please rephrase.

Response 42: Amended lines 103-104.

Chapter Material and Methods

Comment 43: Resampling after 6 months is not mentioned in this chapter (s. subchapter “Follow‐up” and line 268 in the chapter Discussion).

Response 43: We added a sentence where we specified that sampling was repeated, employing the same specifications, after six months in order to assess the evolution of the infection and the success of therapy. Lines 118-121.

Comment 44: Line 78: replace “for a possible” with “for cases of possible”.

Response 44: Amended, line 94.

Comment 45: Lines 83-86: please rephrase for better understanding and readability.

Response 45: Amended, line 103-104.

Comment 46: Line 93: replace “California Mastitis Test (CMT)” with “CMT” (s. comment on line 29).

Response 46: Amended, line 114.

Comment 47: Line 96: replace “on CMT” “in the CMT”.

Response 47: Amended, line 117.

Comment 48: Line 97: add “the” before “National Mastitis Council”.

Response 48: Amended, line 118.

Comment 49: Line 105: replace “with aerobic conditions” with “under aerobic conditions”.

Response 49: Amended, line 129.

Comment 50: Lines 105-107: reliable visual identification of bacteria at species level is not possible. Therefore, rephrase this sentence as follows: Bacterial cultures were inspected visually and growth of one or more uniform colonies were considered relevant and growth of three and more different colonies were considered as contamination.

Response 50: We added the sentence in lines 132-134.

Comment 51: Lines 109-110: remove the sentence “Catalase test was evaluated” and insert “catalase test” behind “Gram stain”.

Response 51: Amended, lines 135-136.

Comment 52: Line 110: remove “serological”

Response 52: Amended, line 137.

Comment 53: Line 112: replaces “doupt” with “doubtful” or “questionable”.

Response 53: Amended, line 138.

Comment 54: Line 114-119 (subchapter “MALDI‐TOF MS methodology”). Several β-haemolytic streptococcal species are difficult to identify properly (Nybakken EJ, Oppegaard O, Gilhuus M, Jensen CS, Mylvaganam H. Identification of Streptococcus dysgalactiae using matrix-assisted laser desorption/ionization-time of flight mass spectrometry; refining the database for improved identification. Diagn Microbiol Infect Dis. 2021 Jan;99(1):115207. doi: 10.1016/j.diagmicrobio.2020.115207. Epub 2020 Sep 22. PMID: 33069003). Therefore the specification of the Bruker database used is relevant. Did you use the the improved version Brukers Compass Library DB-7854?

Response 54: in our laboratory we use MBT Compass reference Library Revision K (version 2022).

Comment 55: Lines 115-116: reword this sentence as follows: Bacterial colonies tested with ambiguous results were subjected to species identification using MALDI-TOF MS. Please consider this issue also in the chapter “Discussion”.)

Response 55: We modified the sentence in lines 147-148 and we added a consideration in the chapter “Discussion” in lines 310-313.

Chapter Results

Comment 56: Data for testing of the dogs is missing (s. lines 260-261).

Response 56: We added a sentence in “Sample collection and bacteriological culture and SCC”, lines 139-144.

Comment 57: Lines 205-219: the subchapter “Biomolecular analysis” describes technical details and should be moved to the chapter “2. Materials and Methods”

Response 57: Thank you for the suggestion but we think that as a matter of fluidity of discussion it is appropriate to leave the position of this paragraph unchanged.

Comment 58: Line 208: Table 1 should be included in the manuscript text with a meaningful legend.

Response 58: We added the table 1 in the text with a meaningful legend. Lines 301-305.

Comment 59: Lines 221-229: The subchapter “Genotypization and genomic characterization” contains technical details and should be moved to the chapter “2. Materials and Methods”.

Response 59: We added a new chapter “Genotyping and genomic characterization” in materials and methods. Lines 178-185.

Chapter Discussion

Comment 60: The isolation of different MLST sequence types in dairy cows, dogs, cats and humans should be discussed in more detail. Thus, please consider the occurrence of the sequence type ST55 in a dairy herd suffering from subclinical mastitis (Eibl C, Baumgartner M, Urbantke V, Sigmund M, Lichtmannsperger K, Wittek T, Spergser J. An Outbreak of Subclinical Mastitis in a Dairy Herd Caused by a Novel Streptococcus canis Sequence Type (ST55). Animals (Basel). 2021 Feb 20;11(2):550. doi: 10.3390/ani11020550. PMID: 33672442; PMCID: PMC7923261; Richards VP, Zadoks RN, Pavinski Bitar PD, Lefébure T, Lang P, Werner B, Tikofsky L, Moroni P, Stanhope MJ. Genome characterization and population genetic structure of the zoonotic pathogen, Streptococcus canis. BMC Microbiol. 2012 Dec 18;12:293. doi: 10.1186/1471-2180-12-293. PMID: 23244770; PMCID: PMC3541175; Pagnossin D, Weir W, Smith A, Fuentes M, Coelho J, Oravcova K. Streptococcus canis genomic epidemiology reveals the potential for zoonotic transfer. Microb Genom. 2023 Mar;9(3):mgen000974. doi: 10.1099/mgen.0.000974. PMID: 37000493; PMCID: PMC10132062.)

Response 60: Thank you for your suggestion. We added some consideration in lines 335-340.

Comment 61: Line 239: replace “coming from” with “originating from”.

Response 61: Amended lines 308.

Comment 62: Line 240: replace “could be cause” by “can cause”, delete “of” and add “a” before “contagious”.

Response 62: Amended line 309.

Comment 63: Lines 241: Please insert a comment on identification of S. canis using MALDI-TOF MS: MALDI-TOF MS is a reliable method for identification and differentiation of β-haemolytic streptococcal species. However, for precise species identification of those bacteria a comprehensive data base is required (Nybakken et al. 2021).

Response 63: We added a comment in lines 310-313.

Comment 64: Lines 118-119: replace “compared with those of known microbial isolates of the commercial library provided by Bruker Daltonics” with “compared with those of the commercial database provided by Bruker Daltonics (Milano, Italia)”.

Response 64: Amended, lines 152-153.

Comment 65: Line 138-142: put the additional information “‐including all major coagulase‐negative staphylococci‐”, “including E. faecalis and E. faecium-” and “‐penicillin‐resistance gene‐” in brackets.

Response 65: Amended, lines 174-176.

Comment 66: Line 143: remove the bracket behind “Prototheca spp.)”.

Response 66: Amended, line 177.

Comment 67: Line 249-251: the meaning of this sentence is unclear. Please rephrase for better understanding.

Response 67: We modified the sentence. Lines 323-325.

Comment 68: Line 250: delete the space after “[28)”.

Response 68: we edited this sentence.

Comment 69: Line 252: remove “can” before “confirm”. Finish the sentence after the word “evidence” and start the next sentence with “In particular”.

Response 69: Amended, line 327.

Comment 70: Line 255: replace “therapy in” with “therapy of”.

Response 70: Amended line 330.

Comment 71: Line 258: replace “being the possible source” with “being a possible persistent source”.

Response 71: Amended, line 333.

Comment 72: Line 260: put “the” before “PubMLST”.

Response 72: Amended, line 335.

Comment 73: Line 261: replace “while cats were impossible to evaluate” with “while this was not possible for cats”.

Response 73: Amended, lines 340-341.

Comment 74: Line 266: replace “understood” with “reported”.

Response 74: Amended, line 346.

Comment 75: Line 267: insert a space between “cases” and “[30]”. Replace “The study” with “This study” or “Our study”.

Response 75: Amended, line 347.

Comment 76: Line 274: replace “cows infected” with “infected cows”.

Response 76: Amended, line 356.

Comment 77: Line 275: replace “the possible” with “a possible” and insert “emphasizing” before “the importance”.

Response 77: Amended, line 356.

Comment 78: Line 276: replace “with a re-examination” with “by re-examination. Remove “of the treated cows for the control of the infection”.

Response 78: Amended, line 358.

Comment 79: Line 279: replace “antimicrobial” with “antimicrobials”.

Response 79: Amended line 361.

Comment 80: Lines 279-283: this sentence is difficult to understand. Please rephrase in shorter sentences.

Response 80: We rephrased the sentence. Lines 362-365.

Comment 81: Line 289: replace “Multi Sequence Typing” with “genotyping using MLST”.

Response 81: Amended, line 372.

Comment 82: Line 294: replace “the improve” with “improvement of” and insert the recommendation “removal of infected long-term shedding in animals”.

Response 82: Amended, line 378.

Comment 83: Line 295: replace “re-infection” with “re-infections”.

Response 82: Amended, line 379.

Comment 84: Line 297: replace “to better understand” with “for better understanding”.

Response 84: Amended, lines 381-382.

Comment 85: Line 298: remove “and to have more accurate about that infections” with “and availability of more accurate data about S. canis infections”.

Response 85: Amended, lines 382-383.

Thank you for your interesting advising and considerations. We also thank you for your careful corrections to the text.

Reviewer 3 Report

Comments and Suggestions for Authors

In this study, Laura Del Sambro and colleagues highlight a case of subclinical mastitis caused by S. canis in a dairy herd, with an increased somatic cell count (SCC). Diagnosis was achieved through conventional bacteriology and Multilocus Sequence Typing. Treatment of infected cows, without optimal hygiene and management measures, proved ineffective. Although the source of infection was not identified, cow-to-cow transmission and long-term infection were observed. To effectively control S. canis in the herd, improved management, early detection, and better milking hygiene are essential. This study contributes to the understanding of subclinical mastitis caused by S. canis, suggesting the need for further research into its epidemiology in dairy cattle. This work is suitable for publication in a veterinary science journal after minor revisions.

Comment: Please rewrite all the references provided in the manuscript in the veterinary science journal format for clarity and consistency.

Author Response

In this study, Laura Del Sambro and colleagues highlight a case of subclinical mastitis caused by S. canis in a dairy herd, with an increased somatic cell count (SCC). Diagnosis was achieved through conventional bacteriology and Multilocus Sequence Typing. Treatment of infected cows, without optimal hygiene and management measures, proved ineffective. Although the source of infection was not identified, cow-to-cow transmission and long-term infection were observed. To effectively control S. canis in the herd, improved management, early detection, and better milking hygiene are essential. This study contributes to the understanding of subclinical mastitis caused by S. canis, suggesting the need for further research into its epidemiology in dairy cattle. This work is suitable for publication in a veterinary science journal after minor revisions.

Comment 86: Please rewrite all the references provided in the manuscript in the veterinary science journal format for clarity and consistency.

Response 86: We rewrited all the references.

Thank you for the suggestion. We are glad that the manuscript may contribute to further epidemiological studies.

Round 2

Reviewer 2 Report

Comments and Suggestions for Authors

Comments of the Reviewer on the manuscript vetsci-3468437-peer-review-v3

General comments

The comments in the first review have been implemented properly.

Nevertheless, questions remain regarding the following general comment from the first review:

„Whole genome sequences are not provided on the platform PubMLST. However, the whole genome sequences of the samples 2, 4, 7, and 21 should be provided in the NCBI genome database. Whole genome sequences are essential for molecular epidemiological studies and represent an important enrichment of the sequence database. A Minimum Spanning Tree (MST) (e.g. goeBURST diagram or cgMLST) including whole genome sequence data from S. canis isolated from different host species like, dairy cows, dogs, cats, and humans would illustrate the results clearly. Furthermore, a minimum spanning tree (MST) based on cgMLST data would characterize S. canis in an unprecedented depth for the first time to the best of my knowledge.“

A reply is requested to the following questions:

  • Whole genome sequences (WGS) from canis isolated from the samples 2, 4, 7, and 21 have been obtained. Will these sequences be made available on the NCBI database as recommended?
  • In the MST only MLST data from isolate 21 is included. Why are the data of the three other samples left out (Figure 1: … the ST24 node, in which isolate from Sample 21 is placed.)?
  • When using WGS data, a MST can be created based on MLST as well as based on cgMLST data. A MST based on MLST data is valuable for comparison with previous studies. However, a MST based on cgMLST might provide a characterization at a much greater depth. cgMLST evaluations are future-oriented and so far unique for canis isolates originating from bovine mastitis. Therefore, characterization of these isolates based on cgMLST would unprecedented and unique.

Clarification of these questions is requested.

Special comment:

Legend to Figure 1: the abbreviations „NT“ and „P“ should be explained.

Author Response

Nevertheless, questions remain regarding the following general comment from the first review:

„Whole genome sequences are not provided on the platform PubMLST. However, the whole genome sequences of the samples 2, 4, 7, and 21 should be provided in the NCBI genome database. Whole genome sequences are essential for molecular epidemiological studies and represent an important enrichment of the sequence database. A Minimum Spanning Tree (MST) (e.g. goeBURST diagram or cgMLST) including whole genome sequence data from S. canis isolated from different host species like, dairy cows, dogs, cats, and humans would illustrate the results clearly. Furthermore, a minimum spanning tree (MST) based on cgMLST data would characterize S. canis in an unprecedented depth for the first time to the best of my knowledge.“

A reply is requested to the following questions:

Comment 1: Whole genome sequences (WGS) from canis isolated from the samples 2, 4, 7, and 21 have been obtained. Will these sequences be made available on the NCBI database as recommended?

Response 1:  We uploaded Sample 21 FASTA assembly sequence together with metadata in NCBI: https://submit.ncbi.nlm.nih.gov/subs/wgs/SUB15152175/overview and https://www.ncbi.nlm.nih.gov/bioproject/PRJNA1231738. See “Data Availability Statement” section for details (line 361-364). Furthermore, we are trying to upload the remaining assemblies into NCBI (Submission Ids: SUB15163527, SUB15163591, SUB15163646); however, assembly sequence analysis is still running.

Comment 2: In the MST only MLST data from isolate 21 is included. Why are the data of the three other samples left out (Figure 1: … the ST24 node, in which isolate from Sample 21 is placed.)?

Response 2: We uploaded the remaining assemblies (Sample 2, Sample 4, Sample 7) into PubMLST resource (submission Id: BIGSdb_20250310075601_2434909_83140). Unfortunately, they necessitate some processing time in order to obtain a PubMLST identifier. Given that they are the same S. canis strain (99.99% average nucleotide identity each other, they will likely have the same ST as Sample 21, so nothing will change within the Figure 1 tree (except ST24 node size).

Comment 3: When using WGS data, a MST can be created based on MLST as well as based on cgMLST data. A MST based on MLST data is valuable for comparison with previous studies. However, a MST based on cgMLST might provide a characterization at a much greater depth. cgMLST evaluations are future-oriented and so far unique for canis isolates originating from bovine mastitis. Therefore, characterization of these isolates based on cgMLST would unprecedented and unique.

Response 3: Thanks a lot for this advice. We searched public databases for S. canis genomes: to date, there are only around 50 public genome assemblies (NCBI), apart from the one described here. We also uploaded the remaining 3 assembly sequences that derive from the same circulating strain.

According to our expertise and pertinent literature, a reliable cgMLST scheme should be based on hundreds of genome assemblies. The ChewBBACA suite (https://chewbbaca.readthedocs.io/en/latest/user/tutorials/chewie_step_by_step.html) requires good quality genomes in order to detect a consistent core gene set and identify alleles. Nonetheless, we will constantly monitor public resources and we will try to generate a preliminary cgMLST scheme, in the event that 80-100 assemblies will be globally present.

Clarification of these questions is requested.

Special comment:

Comment 4: Legend to Figure 1: the abbreviations „NT“ and „P“ should be explained.

Response 4: We explained the abbreviations N, P and NT in line 269,270.

Thanks a lot for the valuable suggestions.